# Weight–Length Relationship Analysis Revealing the Impacts of Multiple Factors on Body Shape of Fish in China

Yushan Li [1], Mingjun Feng [2], Liangliang Huang [1], Peiyu Zhang [3], Hongxia Wang [3], Jingwei Zhang [3], Yuehan Tian [4] and Jun Xu [3,*]

[1] College of Environmental Science and Engineering, Guilin University of Technology, Guilin 541004, China; 2120210467@glut.edu.cn (Y.L.); llhuang@glut.edu.cn (L.H.)

[2] Hubei Provincial Engineering Laboratory for Pond Aquaculture, Engineering Research Center of Green Development for Conventional Aquatic Biological Industry in the Yangtze River Economic Belt, College of Fisheries, Huazhong Agricultural University, Wuhan 430070, China; mjfeng@webmail.hzau.edu.cn

[3] Donghu Experimental Station of Lake Ecosystems, State Key Laboratory of Freshwater Ecology and Biotechnology of China, Institute of Hydrobiology, Chinese Academy of Sciences, Wuhan 430072, China; zhangpeiyu@ihb.ac.cn (P.Z.); hongxiawang@ihb.ac.cn (H.W.); zhangjingwei@ihb.ac.cn (J.Z.)

[4] School of Natural Resources, Faculty of Geographical Science, Beijing Normal University, Beijing 100875, China; 202121051110@mail.bnu.edu.cn

[*] Correspondence: xujun@ihb.ac.cn

**Abstract:** The weight–length relationship (WLR) of fish is a crucial tool in fish biology research and has received extensive attention. However, fish growth is influenced by many factors, and the WLR also changes accordingly. Our aim was to investigate how fish body shape is affected by various factors by analyzing the existing parameters of the relationship between fish body length and weight. We analyzed 198,354 fish specimens belonging to 402 species of 82 families in China and investigated the change of fish body shape in the function of their ecology. Herbivorous fish tended to be shorter and fatter than carnivorous fish, and omnivorous fish fall somewhere in between. This difference could be due to variations in feeding habits and the availability of food sources. Additionally, fish living in lentic waters tended to have a shorter and fatter body shape compared to those living in lotic waters. This could be attributed to differences in swimming behavior in these environments. Furthermore, our results showed that the *b* value decreased as altitude increased, and fish tended to be thinner and longer due to lower oxygen and temperature levels in high-altitude waters. Overall, our study provides valuable insights into the WLR of fish and the impact of multiple factors on fish body shape.

**Keywords:** weight–length relationship; *b* value; fish body shape; multiple factors

**Key Contribution:** In this study, we scrutinized fish data from multiple places in China, then used a generalized linear model to screen out several factors that have a significant impact on the *b* value of the fish WLR. Finally, we analyzed and discussed their impact patterns on the *b* value and body shape of fish.

## 1. Introduction

Fish are the most diverse and numerous vertebrates inhabiting various ecosystems and adapting to different ecological niches in different regions and environments [1]. They occupy a crucial position at the top of the food chain in river ecosystems and their abundance serves as a direct indicator of environmental conditions in water bodies [2]. Furthermore, freshwater fish can influence nutrient cycling, food web structure, energy dynamics, and other functions in the ecosystem, as well as the ecological relationships between water and land [3]. Unfortunately, human activities have resulted in increased water demand, severe environmental degradation, and escalating threats to biodiversity. Scholars have predicted that within the next 25–50 years, 20% of freshwater fish species worldwide will face extinction [1,4].

The weight–length relationship (WLR) is regarded as an important tool for studying the biology of fishes, fish physiology, fish ecology, and stock assessment [5]. First of all, by analyzing the factors that influence an organism's growth, nutritional status, and reproduction, the WLR can provide valuable insights for sustainable management of natural fish populations [6–10]. Furthermore, the WLR can serve as an ecological indicator for assessing the impact of invasive species on local species and their habitats [11]. In addition, the WLR has proven to be useful in estimating the fish condition factor (K), which sheds light on the physiological condition, hypertrophy, and health of the fish [12,13]. Finally, with the help of the WLR, fish length category data can be translated into estimates of biomass in aquatic ecosystems [13] and growth rates [14], which can be used for the comparison of life history and morphology of fish populations across different geographical areas [15]. Therefore, it has significant implications for the protection, management, and utilization of fish populations [16,17].

In fact, the growth of fish and the WLR are affected by various natural and human-related factors, including geographical pattern (e.g., longitude, latitude, and altitude), environmental factors (e.g., temperature, climate, and salinity), biological factors (e.g., sex, habitat, health, and diet), and human activity (e.g., habitat change or destruction, pollution, and fishing pressure) [18–22]. Several studies have extensively investigated the impact of various factors on the WLR, some of which have found differences in the WLR between populations in different geographic areas [23,24]. For instance, Burns (1998) showed that the curve relationship of *Rachycentron canadum* (Rachycentridae) (Linnaeus, 1766) varied between different coastal areas, suggesting that the geographical difference was one of the main reasons for the difference [13]. In addition, Froese (2006), studying the WLR of a large number of fish, underlined differences in fish LWRs due to the effect of differences in sampling season and geographical areas [13]. For example, Wang et al. [25] reported a significant difference in the WLR of *Sebastes schlegelii* (Scorpaenidae) (Hilgendorf, 1880) through seasons, and its *b* value was the lowest in spring. Environmental temperature is another factor that significantly affected the WLR. For instance, a study of six different size classes of *Gadus morhua* (Gadidae) (Linnaeus, 1758) demonstrated that temperature had size-dependent effects on the relative condition factor of all fish, and temperature has the most significant impact on small-sized fish [26]. Additionally, the body shape of fish species has been found to be related to their feeding habits [27]. However, there is a lack of large-scale research and systematic generalization on the relationship between the environmental factors [28] and the WLR. Given the complex interplay between fish weight–length relationship and its influential factors, conducting large-scale studies to comprehensively investigate their relationship becomes imperative.

China plays a critical role as both a major fish producer and exporter [1]. However, in recent decades, significant changes to China's rivers and streams have occurred due to agricultural development, increased demand for drinking water, and the construction of dams. These changes have had a profound impact on the migration of species between different river habitats [29]. Currently, China has recorded around 1323 species of freshwater fish, which account for 9% of the world's total, with 877 of them being endemic to the country [30]. Therefore, this study focuses on the WLR of Chinese fish to explore the impacts of multiple factors on fish body shape through the correlation between the *b* value and fish body shape. Our results will provide a reference for studying the WLR of fish through various environmental factors and contribute to further understanding in the fields of biology, physiology, ecology, biological resource protection, growth, and population dynamics of other species [31].

## 2. Materials and Methods

### 2.1. Fish Data

We scrutinized 71 articles on the WLR of Chinese fish from the Journal of Applied Ichthyology (Table S1). Among them, the sampling area includes 71 sampling points in China (Figure 1), and the sampling time range is 1998–2016. The fish names in the literature

were updated according to FishBase's species list and classification [32]. In case of any inconsistency between fish name, the species classification and population information published by FishBase shall prevail.

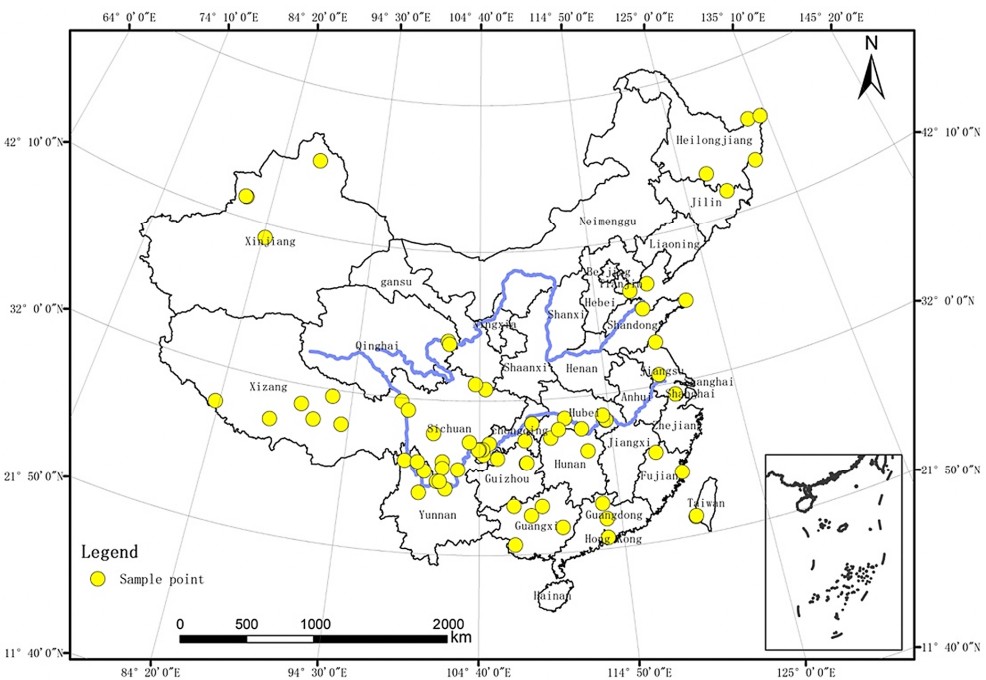

**Figure 1.** Map of China showing sampling locations. The lower right corner shows the South China Sea and its related islands.

Data scrutinized in the literature referred to 198,354 fish specimens, belonging to 402 species of 82 families, through gill nets, bottom trawls, electric fishing, and other methods. The relevant values of *a* and *b* were calculated by the WLR equation.

$$W = a\,L^b \tag{1}$$

The parameter *a* in the equation is the coefficient of the arithmetical WLR, and its 90% values are between 0.001 and 0.050 [13]. When coefficient *b* = 3, the value of *a* represents the form factor of fish, denoted as $a_{3.0}$, which can be used to explore whether the body size of a particular population or species is different from that of other species [13]. Parameter *b* is the exponent of the arithmetical form of the WLR, and 90% of the values ranged from 2.700 to 3.400 [33]. The difference between the *b* value and 3 can indicate the direction and intensity of the change in the morphology or condition of fish [34]. When *b* value is approximately 3, the fish will follow an isometric growth pattern. And when *b* value is significantly different from 3, the fish will follow allometric growth. If *b* > 3, the fish will follow the positive allometric growth, and the fish will often become shorter and fatter [35]. If *b* < 3, fish will follow the negative allometric growth, and will become elongated during the growth process [34].

### 2.2. Influential Factors

We searched the biological information about these fish from FishBase [32]. Fish can be classified into herbivores, carnivores, and omnivores based on their feeding habits. In terms of geographical distribution, the sampling points in this study were categorized into three regions: the southern region, the northern region, and the Tibet region. Based on information from the literature, water environments of sampling points were also classified, with bodies of water that have high mobility, such as rivers and streams, classified as lotic waters (Lotic), and those with low mobility, such as lakes, reservoirs, and bays, classified as lentic waters (Lentic). According to the longitude and latitude of the sampling point, the altitude information of the

place was obtained. Moreover, using the National Oceanic and Atmospheric Administration (NOAA) platform (https://www.ncei.noaa.gov/maps/daily/ (accessed on 17 May 2023)), we obtained atmospheric temperature information for the stations located near the sampling points in the past 50 years (1970–2020), based on the coordinates provided in the literature, and the average values of these parameters were then calculated.

*2.3. Data Analysis*

At the beginning of the data analysis, we observed that the distribution of altitude data had poor normality due to the majority of sampling points being located in low-altitude areas. To address this issue, we transformed the original altitude data using a square root function. Afterward, a generalized linear regression model (GLM) was used to investigate the relationship between the *b* value of the WLR and various factors, including fish feeding habits, geographical region, water body type, altitude, and atmospheric temperature. The optimal model was screened out for analysis using the results of the collinearity test (ensuring that the VIF was less than 5), model stepwise regression (based on the AIC value), and analysis of variance (ANOVA) method. Finally, the "margins" package was utilized to calculate the average coefficient of marginal effects, which could predict the change pattern of the dependent variable when a single independent variable changed. We used the "cplot" function to draw a graph showing the change trend in the *b* value of fish with different feeding habits as the altitude changes. All analyses were conducted using R ver. 4.2.2 [36].

## 3. Results

In this study, we obtained 783 pieces *b* values of fish from 71 articles on the WLR of Chinese fish. Of these data points, the maximum *b* value recorded was 3.910, the minimum was 2.100, and the average was 3.058. Furthermore, Table 1 lists the statistical results of all influential factors.

**Table 1.** Descriptive statistics of influential factors.

|  | Range/Category | Sample Size, *n* | Min of *b* | Mean of *b* | Max of *b* |
|---|---|---|---|---|---|
| | herbivores | 24,044 | 2.540 | 3.100 | 3.510 |
| Diet | omnivores | 60,690 | 2.134 | 3.089 | 3.750 |
| | carnivores | 113,620 | 2.100 | 3.027 | 3.910 |
| Water type | Lentic | 67,439 | 2.100 | 3.099 | 3.910 |
| | Lotic | 130,915 | 2.306 | 3.040 | 3.770 |
| | south | 159,279 | 2.100 | 3.054 | 3.910 |
| Region | north | 28,706 | 2.530 | 3.079 | 3.750 |
| | Tibet | 10,369 | 2.540 | 3.012 | 3.510 |
| Altitude | 0.0–4729.0 (m) | 198,354 | 2.100 | 3.058 | 3.910 |
| Temperature | 0.37–28.31 (°C) | 198,354 | 2.100 | 3.058 | 3.910 |

Collinearity test showed that the variance inflation factor (VIF) of all factors was less than 5, which means there was no significant collinearity between them (Table 2). Region and temperature were deleted due to lack of significant influence on *b* value (Table 3). The results of the generalized linear regression analysis showed that fish feeding habits, water body type, and altitude had a significant impact on the *b* value ($p < 0.01$), with the *p* value for fish feeding habits and altitude being less than 0.001. These results are shown in (Table 4).

**Table 2.** The collinearity test of the model.

| Term | Diet | Water Type | Region | Altitude | Temperature |
|---|---|---|---|---|---|
| VIF | 1.07 | 1.17 | 2.00 | 2.00 | 1.32 |
| VIF 95% CI | (1.02, 1.22) | (1.10, 1.29) | (1.82, 2.24) | (1.81, 2.23) | (1.23, 1.46) |

VIF < 5 indicates low collinearity.

**Table 3.** Model screening by stepwise regression method and ANOVA.

| Model | AIC | Factor | *p*-Value |
|---|---|---|---|
| | −76.27 | Diet | <0.001 |
| | | Water type | 0.020 |
| *b* ~ Diet + Water type + Region + Altitude + Temperature | | Region | 0.076 |
| | | Altitude | 0.001 |
| | | Temperature | 0.098 |
| | −77.86 | Diet | <0.001 |
| *b* ~ Diet + Water type + Altitude | | Water type | 0.008 |
| | | Altitude | <0.001 |

**Table 4.** Summary of generalized linear regression model.

| | Estimate | Std. Error | *p*-Value | Confidence Interval | |
|---|---|---|---|---|---|
| | | | | 2.5% | 97.5% |
| (Intercept) | 3.0891 | 0.0169 | $2 \times 10^{-16}$ | 3.007 | 3.071 |
| Diet herbivores | 0.0993 | 0.0245 | <0.001 | 0.051 | 0.148 |
| Diet omnivores | 0.0654 | 0.0183 | <0.001 | 0.029 | 0.101 |
| Water type Lotic | −0.0500 | 0.0187 | 0.008 | 0.013 | 0.087 |
| Altitude | −0.002 | 0.0005 | <0.001 | −0.003 | −0.001 |

The results of this study indicated that there were differences in the *b* values of fish with different feeding habits. The average *b* values of herbivores, omnivores, and carnivores were 3.100, 3.089, and 3.027, respectively (Table 1 and Figure 2a). Furthermore, the average *b* value (*b* = 3.099) of fish found in lentic water bodies was generally heavier than that found in lotic water bodies (*b* = 3.040) (Table 1 and Figure 2b). Finally, the results showed that the *b* value was negatively correlated with the altitude of the sampling point. As the altitude increased, the *b* value of fish gradually decreased (Figure 2c).

The results obtained by calculating the average coefficient of marginal effects were consistent with the generalized linear regression. We separately tested the relationship between fish feeding habits and fish *b* value when the altitude changed. The results showed that the effect of herbivorous and carnivorous fish on *b* value was significant when the altitude changed, while the effect of omnivorous fish on *b* value was not so significant (*p* = 0.074) (Table 5). It could be observed that the *b* value of herbivorous fish, omnivorous fish, and carnivorous fish decreased as the altitude increased (Figure 3).

**Table 5.** The effect of altitude changes on the *b* value of three different feeding fish species.

| | Diet | Estimate | Std. Error | *p*-Value |
|---|---|---|---|---|
| (Intercept) | herbivores | 3.163 | 0.028 | $<2 \times 10^{-16}$ |
| Altitude | | −0.003 | 0.001 | <0.004 |
| (Intercept) | omnivores | 3.115 | 0.020 | $<2 \times 10^{-16}$ |
| Altitude | | −0.002 | 0.001 | 0.074 |
| (Intercept) | carnivores | 3.067 | 0.017 | $<2 \times 10^{-16}$ |
| Altitude | | −0.003 | 0.001 | 0.001 |

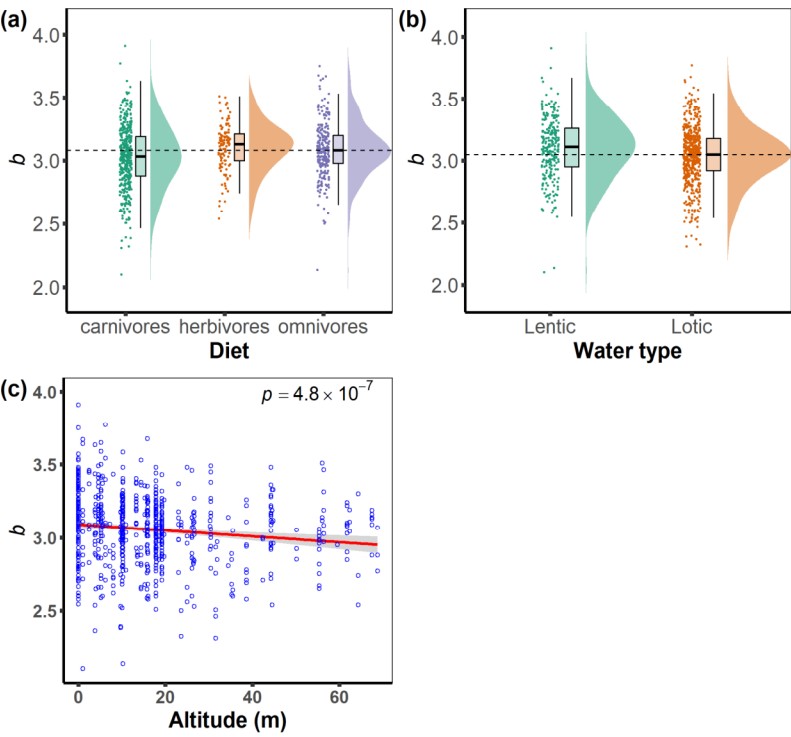

**Figure 2.** Relationship between different factors and *b* value. (**a**) Relationship between feeding habits and *b* value of fish. The black dotted line represents the median of *b* value of omnivorous fish. (**b**) Relationship between water flow type and fish *b* value. The black dotted line represents the median of *b* value of fish in lotic water. (**c**) Relationship between altitude and *b* value of fish. The altitudes were transformed as square root of the original data.

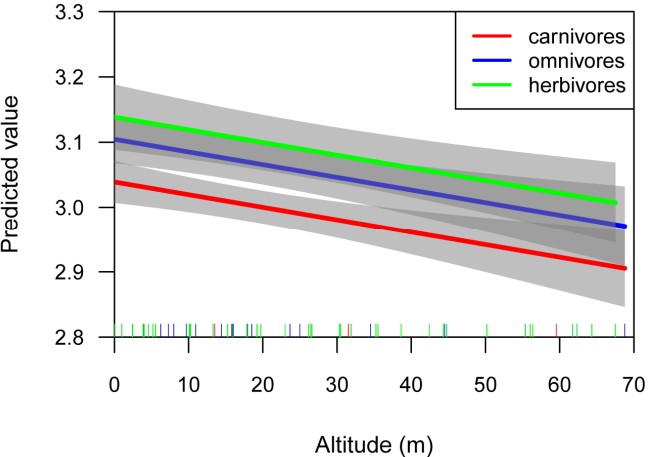

**Figure 3.** The *b* value of fish with different feeding habits changed with the altitude. The altitude data in the figure was the result of the square root of the original data and the shaded part was the confidence interval. The colored bands above the x axis represent the distribution of fish with different feeding habits at altitude.

## 4. Discussion

### 4.1. Effects of Feeding Habits on Fish Body Shape

The study of body morphology in relation to feeding ecology in fish species was pioneered by Keast and Webb in 1966 at a small Canadian lake [37,38]. Subsequently, Catella and Petrere (1998) reported that body shapes of fish species from a floodplain lake in Brazil were related to their feeding habits [27]. In this study, we found that the relationship between the *b* value of fish collected in China was herbivores fish > omnivorous

fish > carnivorous fish, and the body shape of herbivores fish could also be described as wider or shorter than carnivorous fish, being omnivorous fish between them. This is consistent with the results in Bond (1979) that fish with deeper and lateral compressed body feed on the organisms with lower nutrition level in the food web, while fish with slender and longer body feed on the organisms with higher nutrition level in the food web [38]. Similar results were obtained in another research, Gammanpila et al. [39] reported that high trophic level fish exhibited a leptomorphic body shape, whereas low trophic level fish displayed a laterally compressed and deep body shape. Bond (1979) suggested that laterally compressed and short fishes, which are not efficient swimmers but can make quick turns, have body shapes that are likely important for escaping predators [38]. The study of fish growth patterns and their feeding habits also has important applications in aquaculture, The domestication of candidate breeding populations is a crucial step in farming of wild fish, therefore, it is essential to have information about the feeding habits and growth patterns of wild fish to provide appropriate feed and monitor their growth in aquaculture systems [40].

### 4.2. Effects of Water Body Type on Fish Shape

In this study, the average *b* value of fish found in lentic water was generally higher than that found in lotic water, and this suggested that fish found in the former are generally shorter and fatter than those found in the latter. This is parallel with the results of Muchlisin et al. (2015), who found that the Sikundo and Geumpang rivers had a faster flow rate compared to Nagan River, and *b* values of fish in the former rivers were also found lower than that in Nagan River [41]. Water flow is considered a primary driving factor for inducing morphological changes in fish, and the causes of fish body shape variation may be attributed to uncommon hydrological conditions, such as changes in water flow, temperature, and salinity [42]. Dürrani et al. found that *Carassius gibelio* from the Tigris River exhibited a more streamlined body shape compared to those found in natural lakes and reservoirs, and proposed that the purpose of fish body shape variation across different water body types is to minimize energy expenditure [42]. Muchlisin et al. [8] argued that active swimming fish, such as pelagic fishes, tend to have lower *b* values compared to passive swimming fishes, such as benthic fishes. This is likely due to differences in energy allocation for movement and growth between the two types of fish [8]. According to Ahmadi's research [43], deeper-bodied fish were found to prefer lentic water bodies and hiding in surrounding plants to avoid predators. Del Signore et al. [44] found that *Fundulus notatus* living in streams tend to be larger than those inhabiting lakes. Furthermore, at different life stages of fish, there is a positive correlation between water flow velocity and body length [44].

### 4.3. Effects of Altitude on Fish Body Shape

In our study, as the altitude increased, the *b* value of fish gradually decreased (Figure 3). In terms of fish body shape, fish at higher altitudes tended to be thinner and longer, while fish at lower altitudes tended to be shorter and fatter. The influence of altitude on the *b* value of fish might be indirect. Firstly, the *b* value of fish decreased as altitude increased due to the decrease in temperature at higher altitudes. Indeed, there is a positive relationship between temperature and *b* (Figure 4).

Some researchers found that there was a significant difference in the *b* value of the same species of fish in different periods (*b* value of fish in warm period is greater than that in cold period) [45]. This could be explained by the fact that higher temperatures can accelerate the metabolic rate of fish, which, in turn, speeds up the digestion process and promotes faster growth [45]. On the other hand, low water temperatures may lead to insufficient food intake and negative allometric growth in fish [45]. The findings of previous research indicated that warming can increase fish growth rates, but also result in smaller fish at higher temperatures [46–48]. Lindmark et al. reported that the optimal growth temperature for fish decreases as fish size increases [49]. Secondly, oxygen content

is generally more abundant in low-altitude areas, and the metabolic rate of fish is known to be related to oxygen supply [50]. Finally, unique geographical (e.g., slope) and climatic patterns (e.g., precipitation) at different altitudes may give rise to locally adapted fish species that are different from those found in other areas [51]. For instance, the Qinghai–Tibet Plateau region is characterized by an altitude exceeding 5000 m, a cold and arid climate, and harbors mainly cold-water fish species that are adapted to high altitudes and fast-flowing waters (e.g., the endemic fish species *Oxygymnocypris stewartia*, Cyprinidae; and *Nemacheilus subfusca*, Balitoridae). The rivers in the Loess Plateau region exhibit a gradual transition from steep slopes and turbulence to gentleness, and have a dry climate with an average annual temperature of approximately 0 °C, which limits the distribution of most fish species. Consequently, this region is considered one of the most impoverished fish areas, and the fish living in the area exhibit robust ecological adaptability, with the majority of them able to endure low oxygen concentrations and possessing omnivorous feeding habits (e.g., *Triplophysa wuweiensis*, Balitoridae). The southern Chinese landforms comprise three main types, namely, mountains, hills, and plains, and experience a warm or hot climate. In Taiwan Province, significant differences in fish habitats during the dry and rainy seasons have resulted in some endemic species (e.g., *Onychostoma alcorpus*, Cyprinidae; and *Sinogastromyzon puliensis*, Balitoridae) [52]. These distinctive species inhabiting unique geographic and climatic conditions can lead to variations in fish body size across different regions. Overall, the relationship between altitude and the *b* value of fish is complex and can be influenced by various environmental factors, further research is needed to fully understand the mechanisms behind these relationships.

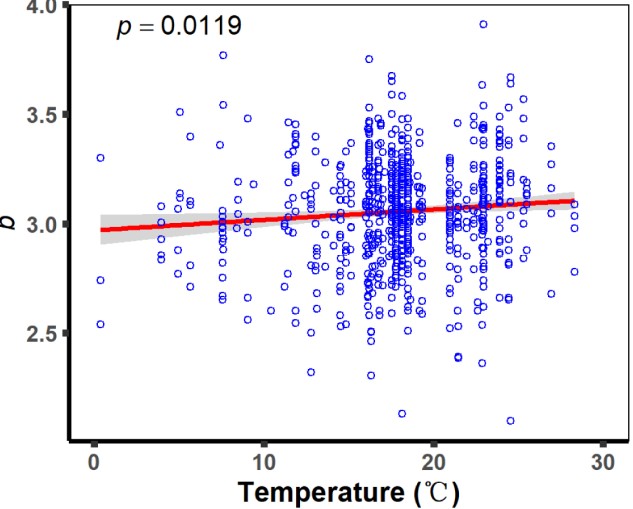

**Figure 4.** Relationship between temperature and *b* value of fish.

## 5. Conclusions

Fish play a critical role in aquatic ecosystems, contributing significantly to material circulation and energy flow [53]. As a result of their ecological importance and visibility, fish are often used as important indicators for evaluating the health of aquatic ecosystems [54]. Monitoring the length, weight, and reproductive behavior of fish can provide insights into their response to environmental changes, enabling the assessment of aquatic ecosystem health [55].

Most of the previous studies on this topic were limited to specific geographical regions, and this study aimed to fill this gap by exploring the relationship between the body shape of fish and multiple factors in a wider range of areas. In particular, data from multiple places in China were analysed by a generalized linear model to screen out factors that have a significant impact on the *b* value of the fish WLR. Results showed that the average *b* value of herbivorous fish was the highest, and the average *b* value of carnivorous fish was the lowest, which meant that herbivorous fish tended to be shorter and fatter than carnivorous one. Furthermore, the average *b* value of fish living in lentic water was higher than that of

fish living in lotic water, indicating that fish living in lentic waters tended to have a shorter and fatter shape than those living in lotic waters. Additionally, the *b* value of fish was negatively correlated with altitude. In terms of fish body shape, fish tended to be thinner and longer with increase of altitude, likely due to lower oxygen and temperature levels in high-altitude waters. The dependency of the WLR of fish on factors such as altitude, water flow, and diet influences the body shape of fish, and provides insights into their biology, physiology, and ecology, and also aids in producing effective strategies for improving protection and management of fish populations.

**Supplementary Materials:** The following supporting information can be downloaded at: https://www.mdpi.com/article/10.3390/fishes8050269/s1, Table S1: fish data.

**Author Contributions:** Conceptualization, J.X.; methodology, J.X., P.Z., M.F., H.W. and Y.L.; software, M.F. and Y.L.; validation, J.X. and Y.L.; formal analysis, P.Z., M.F. and Y.L.; investigation, Y.L., M.F. and J.Z.; resources, Y.L. and Y.T.; data curation, Y.L.; writing—original draft preparation, Y.L.; writing—review and editing, M.F. and J.X.; visualization, Y.L. and M.F.; supervision, J.X.; project administration, J.X. and L.H.; and funding acquisition, J.X. All authors have read and agreed to the published version of the manuscript.

**Funding:** This research was funded by research funds of the Guangxi Key Laboratory of Theory and Technology for Environmental Pollution Control (grant no. 2001K003); the Basic and Applied Basic Research Foundation of Guangdong Province, China (grant no. 2019B1515120065); the National Natural Science Foundation of China (grant no. 31872687); the International Cooperation Project of the Chinese Academy of Sciences (grant no. 152342KYSB20190025); and the National Key Research and Development Program of China (grant no. 2018YFD0900904).

**Institutional Review Board Statement:** No test animals were used during this research.

**Data Availability Statement:** The datasets used during the current study are available from the corresponding author on reasonable request.

**Acknowledgments:** We sincerely thank Yuehan Tian for his assistance in data collation and Guohuan Su for his guidance in writing.

**Conflicts of Interest:** The authors declare no conflict of interest.

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
