# Peer review of "Weight–Length Relationship Analysis Revealing the Impacts of Multiple Factors on Body Shape of Fish in China"

_fishes, doi:10.3390/fishes8050269_

Round 1
Reviewer 1 Report
Present manuscript is satisfactory to be published after minor revisions as follows,
In results portion, as your current work is showing the relationship between the b values with three stations, altitudes, feeding habits, water temperature, water type (standing and flowing), though your table 4 is showing some data of linear regression relationship, but it's not includes both water types, water temperature of three locations of your sampling data, so it's much better that used MS Excel software to show the parameters in the table like a(intercept), b (regression slope), r-values correlation coefficient to show the strength of relationship between the two correlated values in table presentations or graphically presentation, see this published articles below to clearly understand the linear equation as follows,
Simon and Mazlan, Length-Weight and Length-Length Relationships of Archer and Puffer Fish Species.
If possibilities please add latest references about length weight relationship articles in year 2021, 2022
In discussion portion, add the impact of geographical variation of your three stations that are impacting on fish body shape.

Reviewer 2 Report
On the basis of a large amount of empirical data, the authors obtained certain rules, which can be very useful to other researchers.
The manuscript may be accapted with minor technical corrections (see attached file).

Reviewer 3 Report
The study provides important information about LWR is affected by factors such as diet, temperature, watertype, altitude.
93-94: already in results
168-172 + 174-175 + 177-179: move to discussion
181 and 183: Figure 2 (a) and (b) do not show relationship. I suggest to change
189-190: M&M
192-193: Discussions
Please standardize all decimal digits in manuscript (for a and b value)
Table 2: please add number of samples for each range/category
Reviewer 4 Report
The article is overall well-written and organized, providing interesting new insights on the study of LWR-influencing factors. Its originality stems from approaching several influencing factors simultaneously.
An extensive bibliography was consulted, which supports the soundness of the study.
I recommend the publication of this manuscript after minor language proofreading and addressing the specific comments below.
Specific comments
Lines 17 (Abstract): Explain what does “its“ refer to exactly. Fish body shape?
Line 20-21 (Abstract): Avoid repeating the “plump“ body shape, replace it with a synonym.
Line 59: Insert a comma after “addition“.
Line 87: Replace “corrected“ with “updated“.
Lines 93-94: The phrase “Researchers in the literature“ is extremely confusing for the reader. Rephrase the entire construction to “According to the literature investigated... “
Line 97: Put the coefficient “a“ in italics.
Line 100: Write 90% in numbers.
Line 106: Use a capital letter for “If“.
Line 114: “Rivers“ appears twice. What kind on water bodies does the second “rivers“ refer to.
Line 149: 198.354 fish species? This value surely refers to the number of specimens. Correct accordingly.
Line 175: Again, avoid the repetition of “plump“ and find a synonym.
Line 2020-203: Replace “Through introduced“ with “Through the introduction“.
Line 214: Remove the comma after the reference.
Line 230: Remove the comma after the reference.
